# TIGaussian: Disentangle Gaussians for Spatial-Awared Text-Image-3D Alignment

**Jiarun Liu**[1*]  **Qifeng Chen**[1*]  **Yiru Zhao**[1]  **Minghua Liu**[2]  **Baorui Ma**[3]  **Sheng Yang**[1†]

[1]Unmanned Vehicle Dept., Cainiao Inc., Alibaba Group, Hangzhou, China
[2]Hillbot, Sunnyvale, USA
[3]Beijing Academy of Artificial Intelligence, Beijing, China
jiarunliu@zju.edu.cn, {cqf7419, shengyang93fs}@gmail.com

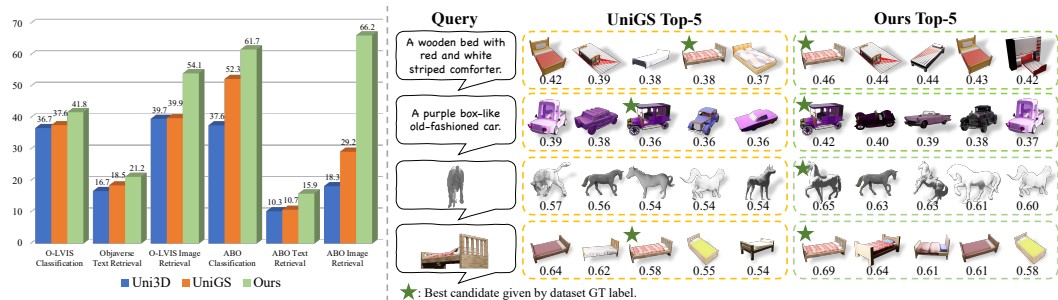

Figure 1: TIGaussian enables 3D modal pretraining on several tasks, e.g., zero-shot classification, text-3D retrieval and image-3D retrieval. **Left:** Compared to prior 3D multi-modal alignment methods – Uni3D and UniGS, TIGaussian presents superior performance on multiple datasets – Objaverse(-LVIS) and ABO. **Right:** In challenging scenarios involving ambiguous or complex queries, TIGaussian demonstrates superior performance owing to its disentangled encoder architecture and specialized cross-modal alignment mechanism. We report the similarity score of each item in the figure.

## Abstract

While visual-language models have profoundly linked features between texts and images, the incorporation of 3D modality data, such as point clouds and 3D Gaussians, further enables pretraining for 3D-related tasks, e.g., cross-modal retrieval, zero-shot classification, and scene recognition. As challenges remain in extracting 3D modal features and bridging the gap between different modalities, we propose TIGaussian, a framework that harnesses 3D Gaussian Splatting (3DGS) characteristics to strengthen cross-modality alignment through multi-branch 3DGS tokenizer and modality-specific 3D feature alignment strategies. Specifically, our multi-branch 3DGS tokenizer decouples the intrinsic properties of 3DGS structures into compact latent representations, enabling more generalizable feature extraction. To further bridge the modality gap, we develop a bidirectional cross-modal alignment strategies: a multi-view feature fusion mechanism that leverages diffusion priors to resolve perspective ambiguity in image-3D alignment, while a text-3D projection module adaptively maps 3D features to text embedding space for better text-3D alignment. Extensive experiments on various datasets demonstrate the state-of-the-art performance of TIGaussian in multiple tasks. Code repository: https://github.com/RUiN-jiarun/TIGaussian.

---

*Equal contribution.
†Corresponding author.

# 1 INTRODUCTION

Recent breakthroughs in vision-language pretraining (Radford et al., 2021; Sun et al., 2023) have established text-image alignment as the foundation for multi-modal understanding. In recent years, significant advances have been made in expanding the field of 3D visual understanding through various 3D representations. Early works in 3D multi-modal alignment primarily relied on *point clouds* processed through 3D convolutions (Zhang et al., 2022; Huang et al., 2023; Xue et al., 2023; 2024) or 3D Vision Transformers (Zeng et al., 2023). Subsequently, works on *meshes* (Song et al., 2023) and *voxels* (Ruan et al., 2024) further explores the operations on organized 3D representations. More recently, UniGS (Li et al., 2025) first leverages *3D Gaussian Splatting* (3DGS) as an advanced 3D representation, demonstrating state-of-the-art text-image-3D alignment across multiple benchmarks.

However, existing methods still facing challenges regarding the abstraction ability of 3D context, as well as the gaps between *text-3D* and *image-3D* modalities. Specifically, our analysis reveals that the current 3DGS-based multi-modal work UniGS, while groundbreaking, suffers from following limitations: (1) *Entangled 3D encoding*: All attributes of each Gaussian primitive are concatenated and encoded as a whole, insufficiently leveraged the distribution pattern and geometric significance of each attribute intrinsic relations between attributes; (2) *Degraded 3D perception*: Forced alignment from a single-view image fails to capture global context, causing a decrease in 3D perception ability.

To overcome these limitations, we propose TIGAUSSIAN, a tri-modal alignment framework that bridges text, image, and 3DGS representations which outperforms existing feature alignment approaches. At its core, our method utilizes multi-branch 3DGS tokenizer to extract 3D contextual information, which reducing inter-attribute coupling, ultimately establishing a compact and effective latent representation for 3DGS. For image-3D alignment, we incorporate a diffusion-enhanced multi-view fusion strategy, which compensates for single-view limitations by leveraging implicit 3D awareness from pretrained diffusion priors. Meanwhile, for text-3D alignment, the encoded feature adapts the 3D latent space through a 3D-text projection module, thereby better matching text embedding structures.

We conduct extensive experiments to validate the performance of TIGAUSSIAN on multiple tasks across the Objaverse (Deitke et al., 2023), ABO (Collins et al., 2022) and SUN RGBD (Song et al., 2015) datasets. As presents in Fig. 1, the results demonstrate the effectiveness of the proposed method, achieving superior performances on zero-shot classification, cross-modal retrieval, few-shot linear probing and open-world scene recognition.

In summary, our contributions can be summarized as follows:

- We propose TIGAUSSIAN, a framework for text-image-3DGS alignment, achieving state-of-the-art performance on various downstream tasks.
- We design a multi-branch 3DGS tokenizer, with specialized Gaussian attribute modeling to enhance 3D feature abstraction and compression.
- We propose a tri-modal alignment strategy through 3D-aware image feature fusion and 3D-text projection module, enabling robust alignment for both text and image modalities.

# 2 RELATED WORK

## 2.1 3D REPRESENTATION LEARNING.

The development of 3D representations is an important factor in promoting research on 3D shape analysis and multi-modality tasks. Early works focused on volumetric grids (Maturana & Scherer, 2015), which enabled structured 3D convolutions but incurred prohibitive computational costs for high-resolution scenes. With the development of neural networks, more works use point clouds as sparse geometric proxies, leveraging permutation-invariant operators like PointNet (Qi et al., 2017a;b) and other classical network architectures (Choy et al., 2019; Ma et al., 2022) to extract localized features. In recent years, there have been many mesh-based feature extraction works in dense 3D representation (Feng et al., 2019; Hu et al., 2022), but the inherent complexity of mesh structures poses significant challenges for effective feature representation. A transformative leap arrived with implicit neural representations like DeepSDF (Park et al., 2019) and NeRF (Mildenhall

et al., 2021), which encode surfaces or radiance fields via coordinate-based MLPs. These methods achieved unprecedented reconstruction quality but suffered from slow optimization and rendering. The recent Gaussian Splatting (Kerbl et al., 2023; Huang et al., 2024) further addresses these limitations by introducing explicit, differentiable primitives optimized for real-time photorealistic rendering. Some recent works exploit the hierarchical structures for scene organization and neural anchors representations (Lu et al., 2024; Ren et al., 2025) for better feature extraction, or embedding the semantic or visual features into 3D Gaussian primitives (Zhou et al., 2024b; Shi et al., 2024; Qin et al., 2024). These works in 3D representation learning in recent years have demonstrated the superiority of 3DGS in scene representation, understanding, and other aspects.

## 2.2 MULTI-MODAL PRETRAINING IN 3D TASKS.

**Text-2D Multi-modal Learning.** Multi-modal pretraining via contrastive learning has revolutionized cross-modal understanding, with foundational frameworks like CLIP (Radford et al., 2021) and EVA-CLIP (Sun et al., 2023) demonstrating the power of aligning image-text pairs. BLIP series (Li et al., 2022; 2023) further enhance the text-image alignment capability through richer data and hybrid encoder-decoder structures. More recent works involve Large Language Models (LLMs) for visual-text understanding (Team, 2023; Zhang et al., 2023; Dong et al., 2024; Chen et al., 2024).

**Text-image-3D Tri-modal Learning.** Recently, more works have extended to 3D fields. Early 3D adaptation strategies such as PointCLIP (Zhang et al., 2022) and CLIP2Point (Huang et al., 2023) circumvented geometric complexity by rendering depth maps from point clouds, effectively projecting 3D data into 2D subspaces for compatibility with pretrained image encoders. Recent efforts like ULIP series (Xue et al., 2023; 2024) and CLIP$^2$ (Zeng et al., 2023) directly encode raw point clouds through 3D transformers, achieving improved robustness in real-world indoor/outdoor benchmarks. Other works such as TriCoLo (Ruan et al., 2024) explore the performance of voxel representations in multi-modal alignment, but are limited to the scale of data and models. OpenShape (Liu et al., 2023) leverages Point-BERT (Yu et al., 2022) and a 3D convolution architecture to train a point cloud encoder for cross-modal understanding. The concurrent work Uni3D (Zhou et al., 2024a) utilizes the Vision Transformer architecture (Dosovitskiy et al., 2021) and large amounts of point cloud data to achieve a model with 1B parameters large model, and realize a state-of-the-art solution for text-image-3D alignment. More recent works (Wang et al., 2025) have adapted Uni3D as the foundation model for 3D learning. To expand into more diverse 3D representations, UniGS (Li et al., 2025) pioneers 3DGS for multi-modal tasks by distilling the Uni3D pretrained model, which makes significant progress. However, these methods suffer from entangled attribute encoding and single-view bias during alignment. Although recent works attempt to encode 3D objects using multi-view images (Lee et al., 2025; Zhou et al., 2025), this inevitably increases the amount of content required for encoding and loses the simplicity of encoding directly for 3D models. Our framework addresses these limitations through an attribute-disentangled 3DGS encoding scheme coupled with cross-view attention mechanisms, dynamically aligned via a pre-projection module that warps the latent 3D manifold to match text-image contrastive embedding spaces.

## 3 METHODOLOGY

### 3.1 OVERVIEW

As illustrated in Fig. 2, with the vanilla 3DGS (Kerbl et al., 2023) as the most prevalent input Gaussian representation, TIGAUSSIAN processes **tri-modal** inputs through three coordinated components: (1) For the **3D** modal, we use a multi-branch 3DGS tokenizer which decomposes geometric and appearance attributes to construct a structured latent embedding $F_{\mathbb{G}}^I$ for spatial context; (2) For the **image** modal, we use a diffusion-enhanced multi-view fusion module for generating 3D-aware visual features $F_{\mathbb{I}}^{mv}$ which aggregates implicit 3D priors from pretrained diffusion models; (3) For aligning the **text** modal, we further project the spatial features $F_{\mathbb{G}}^I$ into $F_{\mathbb{G}}^T$ through a 3D-text projector to ensure modality-invariant feature consistency with text embeddings.

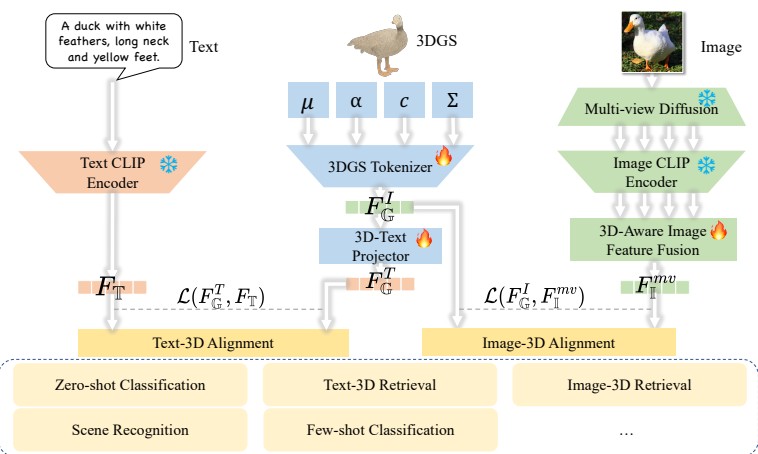

Figure 2: The TIGAUSSIAN framework for text-image-3D tri-modal representation learning. (1) **3DGS Tokenizer**: A multi-branch lightweight network decomposes input Gaussians into separate attributes, generating structured latent embedding $F_{\mathbb{G}}^I$. (2) **Image Modality**: Multi-view diffusion generates consistent views processed through CLIP to produce 3D-aware visual features $F_{\mathbb{I}}^{mv}$. (3) **Text Modality**: The 3D features are projected to text space $F_{\mathbb{G}}^T$ via a learnable projector. Dual contrastive losses $\mathcal{L}(F_{\mathbb{G}}^I, F_{\mathbb{I}}^{mv})$ and $\mathcal{L}(F_{\mathbb{G}}^T, F_{\mathbb{T}})$ align all modalities in a shared embedding space, enabling diverse cross-modal applications.

## 3.2 PRELIMINARIES

**3D Gaussian Splatting.** 3DGS characterizes the object through a collection of anisotropic Gaussian primitives defined within the 3D space (Kerbl et al., 2023). For each 3D Gaussian $\mathcal{G}$, it is described by the following attributes: a center position vector $\boldsymbol{\mu} \in \mathbb{R}^3$, an opacity parameter $\alpha \in [0, 1]$, a set of spherical harmonics (SH) coefficients $\text{SH} \in \mathbb{R}^k$ and a covariance matrix $\boldsymbol{\Sigma} \in \mathbb{R}^{3 \times 3}$:

$$\mathcal{G}(\mathbf{x}) = \exp(-\frac{1}{2}(\mathbf{x} - \boldsymbol{\mu})^\top \boldsymbol{\Sigma}^{-1}(\mathbf{x} - \boldsymbol{\mu})), \tag{1}$$

among which, the SH coefficients can be derived to RGB color $\boldsymbol{c} \in \mathbb{R}^3$, while the covariance $\boldsymbol{\Sigma}$ can be decomposed into scaling factor $\boldsymbol{s} \in \mathbb{R}^3$ and rotation quaternion parameter $\boldsymbol{q} \in \mathbb{R}^4$ as:

$$\boldsymbol{\Sigma} = \mathbf{R}\mathbf{S}\mathbf{S}^\top\mathbf{R}^\top, \tag{2}$$

where $\mathbf{S} \in \mathbb{R}^{3 \times 3}$ is the scaling matrix derived from $\boldsymbol{s}$ and $\mathbf{R} \in \mathbb{R}^{3 \times 3}$ is the rotation matrix derived from $\boldsymbol{q}$.

**Multi-modal Alignment via Contrastive Learning.** After widespread validation in multi-modal works in recent years (Radford et al., 2021; Li et al., 2022; Xue et al., 2023), cross-modal feature alignment through contrastive learning is a highly effective training method. The contrastive loss of two features $\mathcal{L}(F_1, F_2)$ is calculated by the InfoNCE loss (Oord et al., 2018), as:

$$\mathcal{L}(F_1, F_2) = -\mathbb{E}_x\left( \log \frac{\mathcal{L}^+(x, \tau)}{\mathcal{L}^+(x, \tau) + \sum_{j=1}^M \mathcal{L}^-(x, \tau)} \right),$$
$$\mathcal{L}^\otimes(x, \tau) \triangleq \exp(F_1(x) \cdot F_2(x^\otimes)/\tau), \quad \otimes \in \{+, -\}, \tag{3}$$

where $(x, x^+)$ denotes the positive pair of a sample $x$, and $M$ is the number of the negative pairs $\{(x, x^-)\}_{j=1}^M$. $\tau$ is the learnable temperature parameter scaling the similarity scores in contrastive learning. As proved in most 3D multi-modal tasks (Zhou et al., 2024a; Li et al., 2025), the text-image model is pre-aligned (Dosovitskiy et al., 2021; Sun et al., 2023), thus we only need to separately calculate the contrastive loss between 3D features $F_{3\mathbb{D}}$ and pre-aligned image features $F_{\mathbb{I}}$ and text features $F_{\mathbb{T}}$. Therefore, the optimization object can be described as:

$$\mathcal{L} = \lambda_{\mathbb{T}}\mathcal{L}(F_{3\mathbb{D}}, F_{\mathbb{T}}) + \lambda_{\mathbb{I}}\mathcal{L}(F_{3\mathbb{D}}, F_{\mathbb{I}}), \tag{4}$$

where $\lambda_{\mathbb{T}}, \lambda_{\mathbb{I}}$ are balance factors.

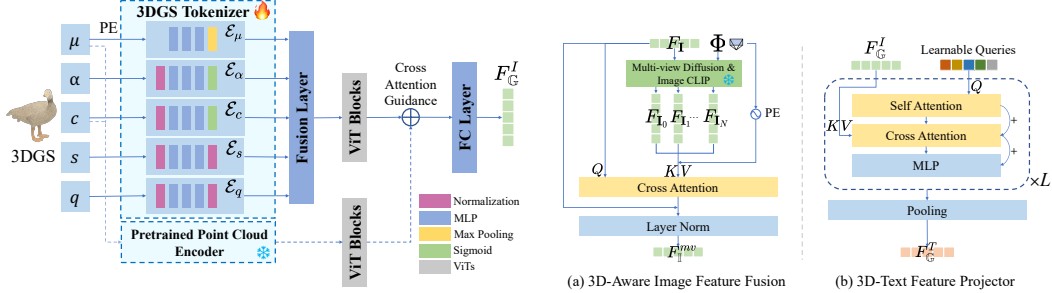

Figure 3: Illustration of 3DGS tokenizer, where multiple branches are tailored for different attributes for a more compact and effective extraction.

Figure 4: Illustration of 3D-aware image feature fusion module and 3D-text projector module.

## 3.3 MULTI-BRANCH 3DGS TOKENIZER

Given a 3D object represented by Gaussians, we first follow existing approaches (Zhou et al., 2024a; Li et al., 2025) to downsample it to 1024 Gaussians by Farthest Point Sampling (FPS) and group them by k-Nearest Neighbor (kNN) algorithm to form several local Gaussian patches of fixed numbers of Gaussians. These local groups serve as processing units for the final ViT-based encoding stage, allowing the model to capture both fine-grained local patterns and global structural relationships. Different from UniGS (Li et al., 2025) which concatenates all Gaussian attributes as homogeneous features, we utilize a well-designed tokenzier to explicitly model the distinct characterization and distributional properties of Gaussian. Specifically, as shown in Fig. 3, each attribute of a Gaussian patch $G = \{\mu, \alpha, c, s, q\}$ – including spatial ($\mu$), appearance ($\alpha, c$) and morphological ($s, q$) components with inherent inter-attribute relationships – is separately fed into its corresponding encoding branch $\{\mathcal{E}_\mu, \mathcal{E}_\alpha, \mathcal{E}_c, \mathcal{E}_s, \mathcal{E}_q\}$. From an information representation perspective (Tishby et al., 2000), such a disentangled token scheme reduces mutual interference among heterogeneous attributes during abstraction and compression. Each branch can adaptively compress its input under an attribute-specific information bottleneck, preserving only the most task-relevant signals. This avoids entangled representations that arise when disparate modalities (e.g., position and color, which possess distinct numerical ranges and representational purposes) are forced to be transformed within a shared domain, thereby resulting in information loss and suboptimal feature alignment. As shown in the example of the bed in Fig. 1, previous methods may ignore the detailed information of "white striped comforter", while our method can highlight this feature.

Each encode branch employs a three-layer MLP for token extraction. For **spatial tokenizer** $\mathcal{E}_\mu$, we adopt an encoder architecture inspired by those used for point clouds (Qi et al., 2017a), incorporating an additional learnable positional embedding module to enhance the extraction of high-dimensional spatial context. Furthermore, a max pooling layer aggregates all point-wise features into a global descriptor by taking the channel-wise maximum values, ensuring permutation invariance and capturing the most salient geometric patterns. For **appearance tokenizer** $\mathcal{E}_\alpha, \mathcal{E}_c$, we employ the sigmoid activation function to model the nonlinear variations in the appearance space and to constrain the value range. As for **morphology tokenizer** $\mathcal{E}_s, \mathcal{E}_q$, we simply use the normalization layer to standardize the output. Finally, the spatial, appearance, and morphological features are concatenated and fused via a two-layer MLP, yielding a unified 3DGS token that holistically captures attribute interdependencies for downstream tasks.

Inspired by UniGS's training strategy, our 3DGS token is prompted via cross-attention guidance from a pretrained point cloud model (Zhou et al., 2024a). Specifically, position $\mu$ and color $c$ attributes are encoded as point cloud features, which are then refined through ViT blocks with cross-attention to integrate pretrained knowledge into the 3DGS token. Finally, the token is processed by a fully-connected layer (FC layer) to adjust its dimension to $d = 512$ and output compact high-dimensional 3D features $F_\mathbb{G}^I \in \mathbb{R}^d$.

### 3.4 Image-3D Alignment

Conventional image-3D alignment methodologies (Xue et al., 2023; Zeng et al., 2023; Zhou et al., 2024a; Liu et al., 2023; Li et al., 2025) predominantly focus on direct alignment between randomly selected single-view image and 3D features. However, this approach inherently compromises the perception capability of 3D features. Specifically, the alignment process constrained to a single perspective may lead to suboptimal matching performance for other viewpoints, as the cross-view consistency of 3D feature representations is not adequately preserved during the alignment procedure. While recent approaches attempt to leverage multiple text-image-3D triplets for a single 3D object for alignment (e.g., ULIP-2 (Xue et al., 2024)) or represent 3D objects directly using 2D multi-view images (e.g., Duoduo-CLIP (Lee et al., 2025)), these methods inevitably compromise either computational efficiency (due to expanded training data requirements) or the benefits of concise explicit 3D representations. To address this issue, we propose a 3D-aware alignment scheme to bridge the gap between image features with 3D features.

As shown in Fig. 4(a), we leverage the multi-view diffusion (Yang et al., 2024) prior to elevate the image information from a single perspective to a 3D spatial dimension. For a single-view image $\mathbf{I} \in \mathbb{R}^{H \times W}$, we first generate $N$ multi-view images by the pretrained diffusion model:

$$\mathcal{D}(\mathbf{I}, \Phi) = \{\mathbf{I}_0, \mathbf{I}_1, \cdots, \mathbf{I}_N\}, \text{where } \Phi = \{\phi_0, \phi_1, \cdots, \phi_N\}, \tag{5}$$

with $\phi_i$ as the $i$-th preset camera angles. Afterwards, each image is fed into CLIP module (Radford et al., 2021) to get the corresponding embedding:

$$F_{\mathbf{I}} = \text{CLIP}(\mathbf{I}) \in \mathbb{R}^d, \quad F_{\mathbf{I}_i} = \text{CLIP}(\mathbf{I}_i) \in \mathbb{R}^d. \tag{6}$$

We then use a perspective-aware cross-attention module for multi-view feature fusion:

$$\text{Attn}(Q, K, V) = \text{Softmax}(\frac{QK^\top}{\sqrt{d}})V, \tag{7}$$

where $Q = F_{\mathbf{I}}$, $K = V = [F_{\mathbf{I}_0}|F_{\mathbf{I}_1}|\cdots|F_{\mathbf{I}_N}] + \text{PE}(\Phi)$, which is the concatenation of all single-view features. $\text{PE}(\cdot)$ stands for sinusoidal positional encoding (Vaswani et al., 2017). The fused 3D-aware feature is:

$$F_{\mathbb{I}}^{mv} = \text{LayerNorm}(F_{\mathbf{I}} + \text{Attn}(Q, K, V)) \in \mathbb{R}^d. \tag{8}$$

During alignment, we replace the image feature $F_{\mathbb{I}}$ in Eq. 4 with $F_{\mathbb{I}}^{mv}$. In this way, 3DGS features will have multi-perspective perception capabilities, thereby enhancing the encoding ability in 3D space.

### 3.5 Text-3D Alignment

Although the current 3D features $F_{\mathbb{G}}^I$ has been aligned with multi-view image features, it is still not sufficient to produce good alignment with text modalities. In order to further narrow the gap between 3DGS and text modalities, we propose a feature projection module that aligns the latent space of 3DGS features and text features, thereby reducing the difficulty of feature alignment. Inspired by previous methods (Li et al., 2023; Sirnam et al., 2024; Hadgi et al., 2025), we utilize a query transformer architecture for latent space projection.

As shown in Fig. 4(b), we employ a set of learnable queries $F_q \in \mathbb{R}^{N_q \times d}$ as soft prompts to iteratively refine the 3D features through an $L$-layer transformer architecture. Each Transformer block consists of three modules: (1) self-attention for query refinement, (2) cross-attention to integrate 3D feature context, and (3) an MLP layer as the feed-forward network. These modules are connected via residual connections.

In detail, the transformer blocks process query token $F_q$ through $L$ sequential layers, where each layer $l \in [1, L]$ performs the following operations:

$$\begin{aligned}
F_q^l &= \text{LayerNorm}\big(F_q^{l-1} + \text{SelfAttn}(F_q^{l-1})\big), \\
F_q^l &= \text{LayerNorm}\big(F_q^l + \text{Attn}(F_q^l, F_{\mathbb{G}}^I, F_{\mathbb{G}}^I)\big), \\
F_q^l &= \text{LayerNorm}\big(F_q^l + \text{MLP}(F_q^l)\big),
\end{aligned} \tag{9}$$

with initialization $F_q^0 = F_q$. Finally, the refined queries are flattened and passed through a pooling layer to produce the compact embedding $F_{\mathbb{G}}^T \in \mathbb{R}^d$ for text alignment.

In summary, the overall loss function for contrastive learning can be specialized to:

$$\mathcal{L} = \lambda_{\mathbb{T}}\mathcal{L}(F_{\mathbb{G}}^T, F_{\mathbb{T}}) + \lambda_{\mathbb{I}}\mathcal{L}(F_{\mathbb{G}}^I, F_{\mathbb{I}}^{mv}), \tag{10}$$

which aims to align features preprocessed for different modalities and bridges the gap between them. Since our work focuses on 3D modality alignment and existing methods (Sun et al., 2023; Li et al., 2023) have effectively addressed multi-modal tasks between text and images, we intentionally exclude the contrastive loss between text features and fused image features in our framework.

## 4 EXPERIMENTS

### 4.1 EXPERIMENTAL SETUP

**Datasets.** We conduct our experiments on three public 3D datasets: Objaverse (Deitke et al., 2023), ABO (Collins et al., 2022) and real-world indoor dataset SUN RGBD (Song et al., 2015), which contain 146k, 7.9k and 6.1k objects, respectively. As for multi-view images, we adopt the MVD-std model of Hunyuan3D-v1 (Yang et al., 2024) to generate 6 views in canonical camera poses for each object before training. Please refer to *supplementary materials* for more details.

**Implementation.** We use the Open-CLIP ViT-B-16 model (Radford et al., 2021) as pre-aligned text-image model and Uni3D-S point cloud model (Zhou et al., 2024a) for 3DGS token guidance. We first train TIGAUSSIAN on Objaverse with the AdamW optimizer (Loshchilov & Hutter, 2019) and the learning rate of $1e^{-4}$ for 15 epochs for all downstream tasks. For ABO and SUN RGBD dataset, we quickly finetune the trained model for another 20 epochs to better fit the dataset. All features are set to the same dimension as $d = 512$ for simplicity. Layer number of 3D-text projector is set to $L = 6$. The balance factors $\lambda_{\mathbb{T}}$ and $\lambda_{\mathbb{I}}$ for contrastive learning are both set to 0.5, following the previous works (Zhou et al., 2024a; Li et al., 2025). All models are trained on 4 A100 GPUs, and inference is performed on a single A100 GPU.

**Baselines.** We compare our method with several state-of-the-art 3D multi-modal methods: CLIP[2] (Zeng et al., 2023), Uni3D (Zhou et al., 2024a), UniGS (Li et al., 2025). We directly inherit the experiment results reported in UniGS. We also reimplement ULIP-2 (Xue et al., 2024) and Duoduo-CLIP (Lee et al., 2025) on the same data scales for classification tasks for fair comparisons. We also conduct additional experiments involving more baselines and comparisons, please refer to our *supplementary materials* for more implementation details.

### 4.2 PERFORMANCE COMPARISON

**Zero-shot Classification.** We evaluate the zero-shot classification performance of TIGAUSSIAN on the Objaverse-LVIS and ABO datasets, which contain 318 and 23 categories, respectively. Table 1 presents the Top-1, Top-3, and Top-5 classification accuracy results. The experimental results demonstrate that TIGAUSSIAN consistently outperforms all baseline methods on both datasets. Especially compared to recent 3DGS-based (Li et al., 2025) and multi-view based models (Lee et al., 2025), our method's consistent superior performance demonstrates deeper exploration of 3DGS's representational advantages in our design, leading to enhanced overall perception and cross-modal capabilities.

**Text-3D Retrieval.** We select 1000 objects from Objaverse and ABO dataset for text-3D retrieval following the sampled test list provided by UniGS. Specifically, we simply calculate the cosine similarity of text embedding $F_{\mathbb{T}}$ and text-aligned 3DGS feature $F_{\mathbb{G}}^T$. Tab. 2 shows the Top-1, Top-5 and Top-10 accuracy. Experimental results show that the proposed text projection module combined with multi-branch Gaussian tokenizer can effectively improve the correlation between 3D and text modalities.

**Image-3D Retrieval.** We further evaluate the image-3D retrieval performances in each batch for Objaverse-LVIS and ABO dataset. The results shown in Tab. 2 indicate that TIGAUSSIAN significantly surpasses all current baselines. We attribute it to the adjustment of image features and the integration of 3D information, thus overcoming the limitations of perspective issues between 3D-2D modal alignment.

Table 1: Performances of different methods on zero-shot classification task. We report the average classification accuracy across all categories.

| Methods | Objaverse-LVIS | | | ABO | | | 3D Repr. |
|---|---|---|---|---|---|---|---|
| | Top-1 | Top-3 | Top-5 | Top-1 | Top-3 | Top-5 | |
| CLIP[2] | 12.35 | 24.62 | 32.91 | 22.58 | 43.83 | 54.56 | Point Cloud |
| ULIP-2 | 29.75 | 46.39 | 55.23 | - | - | - | Point Cloud |
| Uni3D | 36.72 | 57.09 | 65.18 | 37.60 | 59.68 | 70.22 | Point Cloud |
| UniGS | 37.64 | 57.62 | 65.57 | 52.33 | 70.27 | 79.38 | 3DGS |
| Duoduo CLIP | 38.05 | 57.79 | 66.70 | 57.82 | 76.08 | 83.46 | Multi-view (non explicit) |
| Ours | **41.76** | **62.68** | **69.15** | **61.70** | **83.16** | **89.79** | 3DGS |

Table 2: Performance comparison on text-3D and image-3D retrieval tasks. Reported metrics are average retrieval accuracy.

| Methods | Image-3D Retrieval | | | Text-3D Retrieval | | | 3D Repr. |
|---|---|---|---|---|---|---|---|
| | Top-1 | Top-3 | Top-5 | Top-1 | Top-5 | Top-10 | |
| | *Objaverse-LVIS / Objaverse* | | | | | | |
| CLIP[2] | 28.83 | 51.43 | 63.57 | 7.40 | 22.20 | 32.50 | PointCloud |
| Uni3D | 39.65 | 60.72 | 70.51 | 16.70 | 37.10 | 48.10 | Point Cloud |
| UniGS | 41.78 | 62.50 | 72.24 | 21.00 | 39.80 | 53.50 | 3DGS |
| Ours | **54.11** | **73.84** | **81.21** | **21.20** | **45.10** | **56.30** | 3DGS |
| | *ABO* | | | | | | |
| CLIP[2] | 15.29 | 31.74 | 42.74 | 7.09 | 24.34 | 38.94 | Point Cloud |
| Uni3D | 18.25 | 35.26 | 45.29 | 10.29 | 29.21 | 43.67 | Point Cloud |
| UniGS | 26.69 | 46.26 | 56.72 | 11.27 | 30.32 | 43.95 | 3DGS |
| Ours | **66.15** | **73.99** | **85.27** | **15.87** | **40.17** | **53.07** | 3DGS |

**Few-shot Linear Probing.** To further evaluate the learning capabilities of our proposed model, we perform the few-shot linear probing following previous works (Liu et al., 2023; Zhou et al., 2024a). Specifically, we freeze the 3DGS tokenizer and only train a linear classifier on few-shot class labels. We conduct few-shot linear probing on Objaverse-LVIS dataset with labeled training samples per class from 1, 2, 4, 8 to 16. We evaluate each model 10 times and report the average accuracy in Fig. 5. Notably, under the same amount of training data, TIGAUSSIAN outperforms the SoTA 3DGS-based method UniGS.

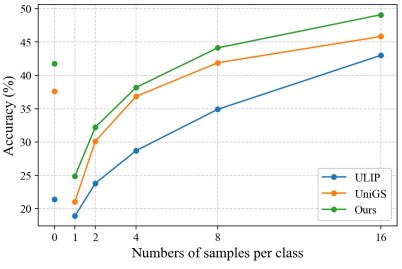

Figure 5: Few-shot linear probing results on Objaverse-LVIS dataset. **0 number of samples per class** stands for zero-shot classification.

| Methods | Mean Accuracy | 3D Representation |
|---|---|---|
| | *SUN RGBD* | |
| CLIP[2] | 41.39 | Point Cloud |
| Uni3D | 61.72 | Point Cloud |
| UniGS | 68.92 | 3DGS |
| Ours | **76.46** | 3DGS |

Table 3: Scene Recognition on SUN RGBD Dataset. We report the average recognition accuracy across all categories. Please refer to *supplementary materials* for detailed results.

**Open-world Scene Recognition.** We follow UniGS (Li et al., 2025) to group the objects in SUN RGBD into 37 categories and take the Top-1 classification accuracy as the scene recognition accuracy. Tab. 3 shows the average accuracy across all categories. The experimental results once again demonstrate the effectiveness of TIGAUSSIAN.

Table 4: Ablation studies on each components. We report the Top 1. accuracy of each task on Objaverse dataset. We abbreviate multi-branch 3DGS tokenizer as **Tkn.**, using multi-view images as **MV.**, multi-view image fusion module as **MVF.**, and 3D-text projector as **TP.**. Moreover, we abbreviate classification as **Cl.**, text retrieval as **TR.**, and image retrieval as **IR.**.

| Exp. | Tkn. | MV. | MVF. | TP. | Cl. | TR. | IR. |
|------|------|-----|------|-----|-------|-------|-------|
| 1 | - | - | - | - | 33.64 | 18.50 | 39.87 |
| 2 | ✓ | - | - | - | 35.57 | 19.15 | 41.68 |
| 3 | ✓ | ✓ | - | - | 37.28 | 19.16 | 46.19 |
| 4 | ✓ | ✓ | ✓ | - | 38.68 | 19.20 | 53.75 |
| 5 | ✓ | - | - | ✓ | 35.71 | 20.80 | 40.52 |
| 6 | - | ✓ | ✓ | ✓ | 37.72 | 17.80 | 52.26 |
| 7 | ✓ | ✓ | ✓ | ✓ | **41.76** | **21.20** | **54.11** |

## 4.3 ABLATION STUDIES

**Ablations on 3DGS Tokenizer.** We first validate the necessity of the multi-branch 3DGS tokenizer which disentangles the Gaussian attributes. Tab. 4 shows the Top-1 accuracy of each task on the Objaverse dataset with different experiment setups. As shown in Tab. 4, the naive UniGS (Exp1) performs less well compared to Exp2, with the 3DGS tokenizer consistently improving accuracy in all tasks. Further analysis (Exp6 vs. Exp7) confirms its role as a crucial 3D context extractor during feature alignment. Removing this component causes significant performance degradation, demonstrating its fundamental importance. These results show that our tokenizer effectively resolves the problem of 3D context abstraction.

**Ablations on Alignment Strategy.** We validate our modality alignment approach through extensive experiments on image and text alignment strategies. Key findings include:

- The **multi-view images feature fusion** (Exp2 vs. Exp4) strategy effectively bridges the image-3D modality gap, significantly improving image retrieval and zero-shot classification.
- **Removing the fusion module** (Exp4 vs. Exp3) leads to noticeable performance degradation, confirming that simply using more training triplets incurs computational costs and doesn't have sufficient performance gains.
- The **3D-text projector** (Exp2 vs. Exp5) successfully aligns 3DGS features with textual representations.

The integrated system (Exp7), combining multi-branch 3DGS tokenizer, 3D-aware image fusion, and 3D-text projector, achieves optimal performance, demonstrating the complementary roles of these modules in cross-modal alignment.

**More Ablation Studies and Discussions.** We also provide detailed discussions and experiments on **guidance strategy**, **scaling-up**, and **generalization**. Please check the *supplementary materials* for more.

## 5 CONCLUSION AND DISCUSSIONS

This paper presents TIGAUSSIAN, a framework that establishes robust alignment among text, image, and 3DGS modalities by addressing three challenges in cross-modal 3D understanding. First, our multi-branch 3DGS tokenizer scheme systematically decouples 3DGS intrinsic attributes, enabling better feature representation of 3D structures. Second, the diffusion-enhanced multi-view fusion mechanism resolves the inherent limitations of single-view observations, ensuring spatial coherence across diverse perspectives and significantly improving the image-3D alignment. Third, the 3D-text projection module effectively bridges the modality gap between continuous 3D feature spaces and discrete text embeddings. Extensive experiments demonstrate state-of-the-art performance in multiple downstream tasks. Our work establishes a new paradigm for leveraging compact latent 3D token in vision-language applications, which further stimulates the potential of 3DGS as an advanced 3D representation in cross-modal understanding.

**Limitations and Future Works.** TIGAUSSIAN has been validated to achieve good results across different datasets and tasks. However, there are two limitations that merit attention. (1) Generalization ability: Although we conduct experiments on Gaussians from various sources and verified the scalability, there may still be scenarios where our method performs degraded on occluded multi objects or real outdoor scenes. (2) Text label dependency: Currently, the quality of text-3D alignment fundamentally relies on training labels. Current benchmarks primarily utilize LLM-generated annotations, which can introduce bias. Future work could explore hybrid supervision paradigms that combine the efficiency of LLMs with expert data label curation for better fine-grained text retrieval.

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
