# Supplementary Materials of TIGaussian: Disentangle Gaussians for Spatial-Awared Text-Image-3D Alignment

**Jiarun Liu**[1][*] **Qifeng Chen**[1][*] **Yiru Zhao**[1] **Minghua Liu**[2] **Baorui Ma**[3] **Sheng Yang**[1][†]

[1]Unmanned Vehicle Dept., Cainiao Inc., Alibaba Group, Hangzhou, China
[2]Hillbot, Sunnyvale, USA
[3]Beijing Academy of Artificial Intelligence, Beijing, China
`jiarunliu@zju.edu.cn`, `{cqf7419, shengyang93fs}@gmail.com`

## A  Implementation Details

### A.1  Dataset Details

For each dataset (i.e., Objaverse (Deitke et al., 2023), ABO (Collins et al., 2022) and SUN RGBD (Song et al., 2015)), we prepare the 3DGS models following methods by UniGS (Li et al., 2025), and sample 1024 Gaussian primitives for each object except for SUN RGBD dataset. As for the text annotations and of each object, we use InternLM-Xcomposer (Dong et al., 2024) to auto-generate the text prompt. To ensure the fairness of comparison, we follow UniGS (Li et al., 2025) for datasets splitting and evalutaion setup. Specifically, we use the same test sets for all dataset and Objaverse-LVIS for evaluation.

### A.2  Training Details

We first train TIGaussian for 15 epochs on Objaverse dataset (Deitke et al., 2023) with 100k objects and evaluate on Objaverse-LVIS dataset. The batch size is 24 for training and 80 for testing. For ABO (Collins et al., 2022) and SUN RGBD (Song et al., 2015) dataset, we quickly finetune the trained model for another 20 epochs to better fit the dataset.

### A.3  Reimplementation Details of Baselines

**UniGS.**  We inherit the test results of UniGS (Li et al., 2025) from the original UniGS paper. Notice that they **DO NOT** release the pretrained checkpoints, and the re-implemented results are slightly downgraded compared to the public results, thus we strictly follow their reported results for fair comparison. The guidance model is Uni3D-S, which is the same used for TIGaussian. Overall, the performance of UniGS generally surpasses other baseline methods.

**ULIP-2 and Duoduo CLIP.**  We follow the training instruction of ULIP-2 (Xue et al., 2024) and Duoduo CLIP (Lee et al., 2025). As for multi-view images number selected for training, we keep it the same as TIGaussian, i.e., 6 images for each object. Moreover, we use the sampled dataset of Objaverse provided by UniGS to train the model instead of using ensemble dataset for fair comparison. It is worth noting that the training data of ULIP-2 has increased by 6 times directly, resulting in a sharp increase in training time and memory overhead.

**Others.**  We inherit the test results of CLIP[2] (Zeng et al., 2023) and Uni3D (Zhou et al., 2024) from the UniGS paper. As for few-shot linear probing result of ULIP (Xue et al., 2023) shows in the main text, we follow the result reported in Uni3D (Zhou et al., 2024).

---

[*]Equal contribution.
[†]Corresponding author.

# B  MORE EXPERIMENTAL RESULTS

## B.1  DETAILED RESULTS OF SCENE RECOGNITION

We report the scene recognition accuracy of 10 typical classes in SUN RGBD dataset in Tab. 1 for detailed comparison with UniGS.

Table 1: Recognition results on SUN RGBD dataset. Notice that the average accuracy is across all 37 categories.

| Methods | Bed | Bookshelf | Chair | Desk | Sofa | Table | Toilet | Bathtub | Dresser | Nightstand | Average |
|---------|-----|-----------|-------|------|------|-------|--------|---------|---------|------------|---------|
| UniGS | 80.57 | 76.49 | 88.65 | 65.45 | 88.61 | 74.23 | **93.45** | 92.06 | 57.64 | 60.16 | 68.92 |
| Ours | **93.87** | **79.33** | **91.62** | **72.62** | **90.56** | **77.50** | 92.82 | **94.67** | **67.50** | **62.15** | **76.46** |

## B.2  COMPREHENSIVE COMPARISONS WITH MORE BASELINE METHODS

We conduct additional baseline comparison on OpenShape (Liu et al., 2023a), TAMM (Zhang et al., 2024), MixCon3D (Gao et al., 2024), Recon (Qi et al., 2023) and ReCon++ (Qi et al., 2024). To ensure fair comparison, we reimplement these baselines by training on Objaverse 100k dataset, and also scaling up our TIGAUSSIAN to train on Objaverse 800k dataset without Objaverse-LVIS.

Notice that since most of the baselines leverage an ensembled point cloud dataset (including Objaverse-LVIS evaluation dataset) for training, which has about 876k instances with 10000 points per object, while the results reported in our paper are conducted on only 100k training instances with 1024 points per object. The comprehensive results shown in Tab. 2 and Tab. 3 reveal the superior performance of our proposed method, demonstrating the effectiveness of 3DGS representations and the proposed tokenization and alignment strategies.

Table 2: Comparison of object classification task on Objaverse-LVIS datset.

| Methods | Base Model | Training Dataset | GS Points | Top-1 | Top-3 | Top-5 |
|---------|-----------|------------------|-----------|-------|-------|-------|
| *Objaverse-LVIS* | | | | | | |
| Recon | PointBERT | | 10000 | 25.1 | 45.6 | 50.1 |
| TAMM | PointBERT | | 10000 | 22.7 | 40.1 | 48.5 |
| MixCon3D | PointBERT | | 10000 | 32.3 | 52.5 | 61.5 |
| ReCon++ | ViT-S | Objaverse-100k | 10000 | 26.9 | 47.5 | 53.6 |
| ReCon++ | ViT-B | | 10000 | 31.0 | 50.5 | 55.2 |
| Ours-S | Uni3D-S | | 1024 | 41.8 | 62.7 | 69.1 |
| Ours-S | Uni3D-S | | 10000 | **46.7** | **68.6** | 74.5 |
| Ours-B | Uni3D-B | | 1024 | 46.6 | 67.5 | **75.4** |

Table 3: Comparison of object classification task on Objaverse-LVIS datset with larger training dataset.

| Methods | Base Model | Training Dataset | GS Points | Top-1 | Top-3 | Top-5 |
|---------|-----------|------------------|-----------|-------|-------|-------|
| *Objaverse-LVIS* | | | | | | |
| OpenShape | PointBERT | Ensembled w/o LVIS | 10000 | 39.1 | 60.8 | 68.9 |
| TAMM | PointBERT | Ensembled w/o LVIS | 10000 | 42.0 | 63.6 | 71.7 |
| MixCon3D | PointBERT | Ensembled w/o LVIS | 10000 | 47.5 | 69.6 | 76.2 |
| ReCon++ | ViT-B | Ensembled w/o LVIS | 10000 | 49.6 | 70.2 | 78.4 |
| UniGS | Uni3D-S | Objaverse-800k w/o LVIS | 1024 | 48.8 | 69.1 | 76.9 |
| Ours | Uni3D-S | Objaverse-800k w/o LVIS | 1024 | **50.1** | **73.6** | **79.6** |

## C    ABLATIONS ON GUIDANCE

Compared to raw point cloud, 3DGS has richer structural priors, since each Gaussian contains position, color, scale, rotation and opacity, which enables explicit modeling of local geometry and view-dependent appearance. As we proposed in the paper, the position and color attribute of 3DGS is similar to point cloud, thus the spatial tokenizer could directly benefit from the pre-trained point cloud encoder (e.g., Uni3D (Zhou et al., 2024) with ViT (Dosovitskiy et al., 2021)). Besides, the pre-trained model is trained on much larger point cloud datasets, thus its generalizability could be inherited and make the training process easier. Based on this design, our proposed 3DGS tokenizer fully utilizes other attributes of GS to better perform multimodal alignment. Furthermore, we believe the choice of 3D representation has long-term implications for generative and downstream tasks. While point clouds are simpler, 3DGS bridges the gap between discrete 3D elements and continuous radiance fields — offering a promising direction for future 3D-aware generation and understanding systems. We add additional experiments on training 3DGS encoder without pretrained Uni3D point cloud model guidance. As shown in Tab. 4, leveraging pretrained model has significant advantages.

Table 4: Performances of different methods on zero-shot classification task. We report the average classification accuracy across all categories.

| Methods | Top-1 | Top-3 | Top-5 |
|---|---|---|---|
| *Objaverse-LVIS* | | | |
| Ours w/o pretrained | 27.93 | 47.74 | 57.08 |
| Ours | **41.76** | **62.68** | **69.15** |

## D    ABLATIONS ON MULTI-VIEW DIFFUSION MODULE

We also analysis the impact of using different multi-view diffusion models or ground-truth rendered images. We select 6 multi-view images for Objaverse dataset following several methods: (1) Hunyuan3D-v1-lite (Yang et al., 2024), (2) Hunyuan3D-v1-std, (3) rendered ground-truth images given by Zero-1-to-3 (Liu et al., 2023b). As shown in Tab. 5, the results are slightly different from each other depending on the performance of chosen diffusion model.

Table 5: Performances of our method training on different source of multi-view images on zero-shot classification task. We report the average classification accuracy across all categories.

| Methods | Top-1 | Top-3 | Top-5 |
|---|---|---|---|
| *Objaverse-LVIS* | | | |
| Ours w/ Hunyuan3D-v1-lite | 39.98 | 61.13 | 68.42 |
| Ours w/ Hunyuan3D-v1-std | **41.76** | **62.68** | 69.15 |
| Ours w/ GT | 41.03 | 62.31 | **69.78** |

## E    ABLATIONS ON MULTI-VIEW IMAGE NUMBERS

To estimate the impact of the number of multi-view images involved in TIGAUSSIAN, we conduct an additional ablation study on different numbers of multi-view images as shown in Tab. 6. xperimental results have shown that selecting more multi-view images does not necessarily mean better multimodal alignment. We hypothesize that this is due to over-alignment between the image and 3DGS modalities when too many views are used, causing the fused visual features to become overly specific and thus harder to align with the more abstract text representations. In addition, as the number of perspectives increases, the requirement for the authenticity of multiple perspectives will also increase, and the number of model parameters will also multiply. Experimental data shows that selecting 6 multi view images can achieve a balance.

Table 6: Performances of our method training with different numbers of multi-view images.

| # MV Images | Classification Top-1 | Image Retrieval Top-1 |
|---|---|---|
| *Objaverse-LVIS* | | |
| 1 | 35.71 | 40.52 |
| 3 | 39.61 | 46.79 |
| 6 | **41.76** | 52.11 |
| 12 | 40.52 | **52.37** |

## F  RUNTIME AND COST ANALYSIS

We further compare the inference time cost with baseline methods on Objaverse-LVIS dataset. All time tests are conducted on a single A100 GPU. As shown in Tab. 7, our method strike a balance between time cost and accuracy. Notice that our disentangled encoder does not introduce too much additional overhead while achieving high performance surpassing compared to UniGS. Compared to methods based on multi-view representation (i.e., Duoduo CLIP (Lee et al., 2025)), our inference time has a significant advantage. As for multi-view generation part, we take it as the training overhead, as Hunyuan3D-v1-lite takes 800ms to generate 6 views per image. Yet as we've discussed in Sec. D, when using ground-truth multi-view images provided by original Zero-1-to-3 dataset, there is no significant training overhead involved.

Regarding the model complexity, the GFLOPs of 3DGS Tokenizer is 88.2M, which takes about 80% of the total FLOPs our model. The image fusion module takes 1M for cross attention only, and 3D-text projector takes 24M for multi-layer transformer. The maximum GPU memory of TIGaussian is 32G during inference, which can be easily perfomed on a single RTX4090 GPU.

Table 7: Comparisons of inference computational cost on Objaverse-LVIS dataset.

| Methods | Inference Time (ms) | Top-1 Acc. |
|---|---|---|
| *Objaverse-LVIS* | | |
| Uni3D | 97.96 | 36.72 |
| UniGS | 185.75 | 37.64 |
| Duoduo CLIP | 235.79 | 38.05 |
| Ours | 192.86 | 41.76 |

## G  ABLATIONS ON TEXT CAPTIONS

In the main experiments, we use the InternLM-Xcomposer generated text prompt for text modal alignment. In order to further discuss the impact of text label selections, we conduct additional experiments on text label generated by Cap3D, an advanced text label tools which has been evaluated by human experts. As shown in Tab. 8, the results are slightly different. Shorter text prompts are more likely to have better classification result.

Table 8: Performances of our method training on different source of text prompt on zero-shot classification task. We report the average classification accuracy across all categories.

| Methods | Top-1 | Top-3 | Top-5 |
|---|---|---|---|
| *Objaverse-LVIS* | | | |
| Ours w/ InternLM | **41.76** | **62.68** | **69.15** |
| Ours w/ Cap3D | 40.12 | 60.35 | 67.82 |

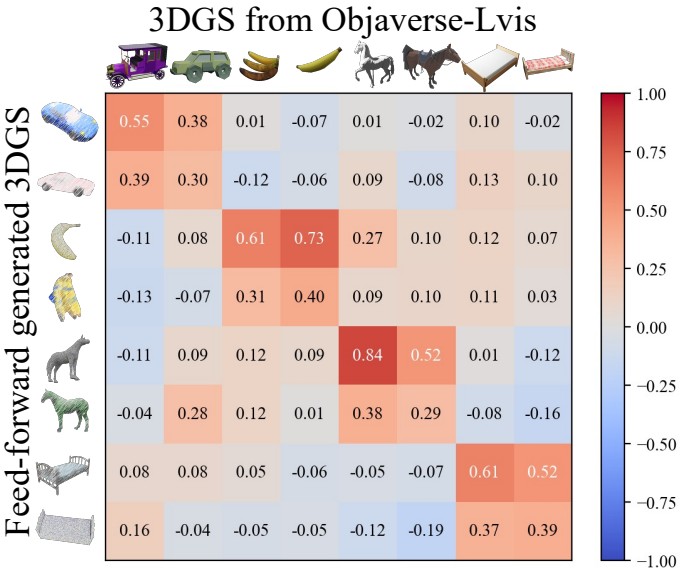

Figure 1: 3DGS feature similarity matrix between Objaverse objects and feed-forward generated 3DGS objects. We select 4 categories from Objaverse-LVIS (i.e., cars, bananas, horses and beds), and compute the cosine similarity of the 3D features of original 3DGS (top) and generated 3DGS (left). The diagonal form of the heatmap proves that our 3D features provide relatively high similarity of objects from the same category, despite their different data sources.

## H   ANALYSIS OF GENERALIZATION ABILITY

To demonstrate the generalizability of our proposed method as a 3DGS encoder, we present similarity heatmaps of 3D embeddings from 3DGS objects of various sources. Fig. 1 shows the correlation between original 3DGS objects and the generated 3DGS objects by feed-forward reconstruction method AnySplat (Jiang et al., 2025), which proves that our method is insensitive to the data source of 3DGS and has good generalization ability. Please refer to *supplementary materials* for details.

As for implementation, we leverage the VGGT-based (Wang et al., 2025) feed-forward 3DGS generation method AnySplat (Jiang et al., 2025) to generate several 3DGS objects from the Objaverse-LVIS dataset. Specifically, we randomly select only 3 views of each object and directly get the 3DGS representation results. Fig. 2 shows some visualization results of the rendered image of the generated 3DGS.

## I   MODEL SCALING UP

We test the performances results of the proposed method and the baseline method on different scales of training data. As shown in Fig. 3(a), scaling up the training data of TIGAUSSIAN can effectively improve the performance on various downstream tasks. Meanwhile, our method surpasses the baseline method and demonstrates effective improvement in training with higher data volumes. In order to further demonstrate the upper limit of our proposed method, we used pretrained point cloud models with different parameter numbers (i.e., Uni3D-S, Uni3D-B and Uni3D-L) to train our encoder, result in Ours-S, Ours-B and Ours-L, respectively. Fig. 3(b) shows Top-1 performances of our method using different backbone.

As for the detailed comparison of scaling up on training dataset and sampled 3DGS points, please refer to Tab. 2 and Tab. 3 in Sec. B.2.

GT Image    Rendered Image    GT Image    Rendered Image

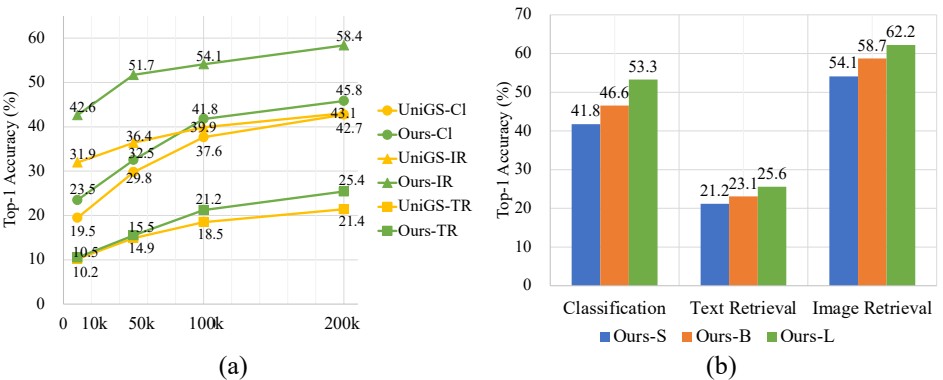

Figure 2: Comparison between the GT images and the rendered images of the 3DGS generated by AnySplat (Jiang et al., 2025).

Figure 3: Analysis on scaling up the model. (a) Top-1 accuracy on Objaverse dataset with different scales of training data. We abbreviate classification as **Cl**, image retrieval as **IR** and text retrieval as **TR**. (b) Top-1 accuracy on Objaverse dataset with different backbone setups.