# OpenReview forum: "TIGaussian: Disentangle Gaussians for Spatial-Awared Text-Image-3D Alignment"
_ICLR.cc/2026/Conference — ICLR 2026 Poster_

### Official Review · Reviewer_yocs · 2025-10-30

**Soundness:** 3
**Presentation:** 3
**Contribution:** 3
**Rating:** 6
**Confidence:** 4

**Summary:**

This paper introduces TIGAUSSIAN, a novel framework for aligning text, image, and 3D modalities, specifically using 3D Gaussian Splatting (3DGS) representations. The core contributions are threefold: 1) a multi-branch 3DGS tokenizer that disentangles and separately encodes geometric, appearance, and morphological attributes of Gaussian primitives to create a more effective latent representation; 2) a diffusion-enhanced multi-view fusion mechanism for image-3D alignment that generates multiple views from a single image to create a more holistic, 3D-aware feature representation, mitigating single-view bias ; and 3) a 3D-text projector module that uses a query transformer to map 3D features into the text embedding space for improved alignment. The authors demonstrate through extensive experiments on datasets like Objaverse, ABO, and SUN RGBD that TIGAUSSIAN achieves state-of-the-art performance on various downstream tasks, including zero-shot classification, text-3D retrieval, and image-3D retrieval.

**Strengths:**

1. **Writing**: This paper is clear and easy to follow.
2. **Cross-Modal Alignment Strategies:** The paper introduces sophisticated strategies for both image-3D and text-3D alignment that address key challenges. The diffusion-enhanced multi-view fusion for image-3D alignment is a clever way to overcome the ambiguity and information loss inherent in single-view representations without requiring actual multi-view data during inference. Similarly, the 3D-text projector is a principled approach to bridge the modality gap, which is a persistent challenge in this area.
3. **Empirical Results:** The method is thoroughly evaluated on multiple benchmarks (Objaverse, ABO, SUN RGBD) and across a variety of tasks (zero-shot classification, cross-modal retrieval, few-shot learning).

**Weaknesses:**

1. **Lack of Large-Scale Dataset Validation:** The primary weakness of this work is the scale of the datasets used for the primary training and validation. While the mentioned Objaverse, with 146k objects, is a respectable size. The current experiments, while strong, do not fully demonstrate the scalability of TIGAUSSIAN on entire Objaverse with 800k objects. Training and evaluating on a significantly larger dataset would provide more convincing evidence of the model's robustness and generalizability.
2. **Comparisons to other methods:** It will be better to report the performance comparisons to ReCon, ReCon++ of ShapeLLM, TAMM on used datasets for completeness.
3. **Complexity and Computational Cost:** The proposed framework introduces several new components. This adds complexity and likely significant computational overhead compared to simpler baseline models. While the performance gains are clear, a discussion on the trade-offs in terms of computational cost, training time, and memory usage would be beneficial. For instance, UniGS report the ablation study on computational cost.

**Questions:**

1. Regarding the scalability: The authors mention that Objaverse contains 146k objects. Have the authors considered or attempted to pre-train TIGAUSSIAN on larger-scale dataset (full objeaverse) to test the limits of the proposed architecture?
2. Regarding the 3DGS tokenizer: The paper mentions using FPS to downsample to 1024 Gaussians. How sensitive is the model's performance to the number of sampled Gaussians? Does the performance saturate at 1024, or could further improvements be gained with more Gaussians at the cost of computation?
3. Regarding the multi-view fusion: The framework generates 6 views using a diffusion model. What was the rationale for choosing 6 views? Is there a performance trade-off with the number of generated views? For instance, would using 3 views significantly degrade performance, or would 12 views provide a notable boost?

---

> ### Author Response · Authors · 2025-11-27
> **Reply to Reviewer yocs**
>
> We sincerely appreciate your valuable feedback. All the experiments and discussions below will be included in the final version of the paper, including citations and discussions of relevant baselines.
>
> Concerning weaknesses:
> 1. **Concerning of scalability.** We conduct experiments on full 800k objaverse datasets. The experiment reveals the strong potential and scalability of TIGaussians.
>
>     Tab.D. Ablation study on training on larger dataset.
>     |           | Base Model | Training Dataset | Sampled Points |   Top-1 | Top-3 | Top-5 |
>     |  ----     | ----       | ----             |   ----         |   ----  | ----  | ----  |
>     | Ours      | Uni3D-S    |  Objaverse 100k  |   1024         |   41.8  |  62.7 | 69.1  |
>     | Ours      | Uni3D-B    |  Objaverse 100k  |   1024         |   46.6  |  67.5 | 75.4  |
>     | Ours      | Uni3D-S    |  Objaverse 800k w/o LVIS |   1024  |   **50.1**  |  **73.6** | **79.6** |
>
> 2. **More baselines.** Please refer to Tab.A and Tab.B in **General Responses**.
> 3. **Discussions of computational cost.** We've already discussed the inference time cost in our supplementary materials (see Sec.E.). Regarding the model complexity, the GFLOPs of 3DGS Tokenizer is 88.2M, which takes about 80% of the total FLOPs our model. The image fusion module takes 1M for cross attention only, and 3D-text projector takes 24M for multi-layer transformer. The maximum GPU memory of TIGaussian is 32G during inference, which can be easily perfomed on a single RTX4090 GPU.
>
> Concerning questions:
> 1. **Question about scalability.** Please refer to response of Weakness 1.
> 2. **Question about more sampled points.** We conduct experiments on more sampled points of GS for training. Notice that most of the point cloud baseline methods sampled 10k points from the original point cloud. We reported the results of 1024 GS points to keep a fair comparison with UniGS in the main paper, since the training process and cost may grow larger as the number of points grows. The conclusion is clear that more GS points can contribute to the model performance, which shows the potential of our method.
>
>     Tab.E. Ablation study on sampled GS points.
>     |           | Base Model | Training Dataset | Sampled Points |   Top-1 | Top-3 | Top-5 |
>     |  ----     | ----       | ----             |   ----         |   ----  | ----  | ----  |
>     | Ours      | Uni3D-S    |  Objaverse 100k  |   1024         |   41.8  |  62.7 | 69.1  |
>     | Ours      | Uni3D-S    |  Objaverse 100k  |   10000         |   **46.7**  |  **68.6** | **74.5**  |
>
> 3. **Question about the number of multi-view images.** We've discussed this question before when replying to reviewer hpbf. For the convenience of reading, we will discuss it again here. We conduct experiments on selection of multi-view images.
>
>     Tab.C. Ablation study on different number of multi-view images.
>     |  Multi-view Number         | Classification Top-1 | Image Retrieval Top-1 |
>     |  ----     | ----       | ----            |
>     | 1 | 35.71  |  40.52  |
>     | 3      | 39.61  |  46.79  |
>     | 6  | **41.76**  |  52.11  |
>     | 12   | 40.52      |  **52.37**  |
>
>     Experimental results indicate that increasing the number of multi-view images does not always lead to better multimodal alignment. Furthermore, as the number of viewpoints increases, the demand for accurate camera poses grows, and the model's computational and parametric complexity rises accordingly. Our experiments show that using 6 multi-view images achieves an effective balance between alignment quality and modeling efficiency.

---

### Official Review · Reviewer_Xq4U · 2025-10-31

**Soundness:** 2
**Presentation:** 2
**Contribution:** 3
**Rating:** 6
**Confidence:** 5

**Summary:**

The paper proposes a framework to train a unified text-image-3D features by decoulping the parameters of 3D gaussian. It introduces a multi-branch 3DGS tokenizer to fuse the 3D features within various perspectives. In addition, it proposes a tri-modal alignment strategy to interact with multi-view generated images. The experiments prove that it achieves impressive improvements across retrieval and scene understand tasks.

**Strengths:**

- The paper achieves impressive improvement across various tasks.

- The writing is easy to understand and mask sense.

**Weaknesses:**

- Lack of comparisons. One of the core contribution is 3D-aware image feature fusion. However, the usage of multi-view rendering images has been proposed in the JM3D [1], which mitigates the contribution. The authors should either discuss the differences or include a comparison.

- Lack of the experiments of perception. Whether JM3D or ULIP has experiments about 3D Res or Object detection to support the ability in sparse perception. The paper needs the similar experiments.

- Contribution. In Tab.4, the comparison between the second and fifth lines shows that the 3D-text projector did not perform well, which impacts the validity of the author's claimed significant contribution.

[1] Beyond First Impressions: Integrating Joint Multi-modal Cues for Comprehensive 3D Representation, MM 23

**Questions:**

- Why use a single image generation method to obtain multi-view images instead of directly rendering from the model?

- What backbone does ULIP2 use in Tab. 1?

- Appendix Table 2 shows that the pre-trained point cloud encoder has a significant impact on the results, which weakens the overall contribution of the method. What is the significance of using 3D Gaussian to compare point clouds?

---

> ### Author Response · Authors · 2025-11-27
> **Reply to Reviewer Xq4U**
>
> We sincerely appreciate your valuable feedback. All the experiments and discussions below will be included in the final version of the paper, including citations and discussions of relevant baselines.
>
> Concerning weaknesses:
> 1. **More baselines and discussions of JM3D.** We've conducted comparison experiments with more baselines in Tab.A and Tab.B in **General Respond**. Regarding the difference between TIGaussians and JM3D, althoug JM3D encodes multi-view images to a combined feature, it does not directly align the feature with the 3D feature extracted from point cloud. Instead, it proposes a joint multi-modal alignment module, which align the text feature and multi-view image feature first. This may lead to the introduction of some errors in the updated features, thereby reducing the performance of 3D feature alignment.  Our method directly focuses on the feature differences between 3D modalities and other modalities, and aligns them accordingly. We'll cite and discuss the related works in the final version.
> 2. **Concerning about perspection ability.** We conduct experiments on object recognition task on SUN RGBD dataset (Tab.3 & Tab.5), which somehow reveal the ability of perception.
> 3. **Clarification of contributions.** The comparison between Exp2 and Exp5 in Tab.4 reveals that the 3D-text projector module can improve the overall performance of 3D object classification and text retrieval. But at the same time, the solely introduction of  3D-text projector makes it bit more difficult to align with image features, so the performance of image retrieval module decreases slightly. The comparison between Exp4 and Exp5 can show the significance of text projector module in text-3D alignment. Furthermore, the alignment of text modalities remains a challenging task in the future, and strategies such as text-image pre alignment in works like JM3D are worth discussing and researching.
>
> Concerning questions:
> 1. **Question about multi-view images source.** It's fine to use the pre-rendered images, while we conduct the ablation study in supplementary (see Sec. D.) However, for some dataset without direct multi-view images or when direct inference with single-view image, we can easily leverage the multi-view diffusion prior to generate multi-view image without additional effort. As for the hallucination problems, our experiment in Sec.D. has proved that there's no much difference between generated images and real images.
> 2. **Question about ULIP-2 backbone.** We use PointBERT backbone for ULIP-2, which is reported corresponding to their best perfromance.
> 3. **Question about pre-trained point cloud encoder.** Compared to raw point cloud, 3DGS has richer structural priors, since each Gaussian contains position, color, scale, rotation and opacity, which enables explicit modeling of local geometry and view-dependent appearance. As we proposed in the paper, the position and color attribute of 3DGS is similar to point cloud, thus the spatial tokenizer could directly benefit from the pre-trained point cloud encoder (e.g., Uni3D with ViT). Besides, the pre-trained model is trained on much larger point cloud datasets, thus its generalizability could be inherited and make the training process easier. Based on this design, our proposed 3DGS tokenizer fully utilizes other attributes of GS to better perform multimodal alignment.
> Furthermore, we believe the choice of 3D representation has long-term implications for generative and downstream tasks. While point clouds are simpler, 3DGS bridges the gap between discrete 3D elements and continuous radiance fields — offering a promising direction for future 3D-aware generation and understanding systems.
> Meanwhile, we conduct an extensional experiment of unfreeze the pretrained Uni3D backbone. The results indicate that the overall performance is gaining through this architecture design.

---

### Official Review · Reviewer_hpbf · 2025-10-31

**Soundness:** 3
**Presentation:** 3
**Contribution:** 3
**Rating:** 6
**Confidence:** 3

**Summary:**

TIGAUSSIAN, a framework for text-image-3D Gaussian (3DGS) multimodal alignment, is proposed in this research. The main contribution is to solve the problems of single-view bias and attribute entanglement encoding that exist in current multimodal approaches for 3D settings. To make up for the shortcomings in single-view 3D sceneries, they have developed a diffusion-enhanced multi-view fusion module and a multi-branch 3DGS tokenizer module that decouples from the spatial, appearance, and morphological aspects of 3DGS. Finally, an engineered 3D-text projection module is used to align 3D features with text features. Tests on datasets including Objaverse, ABO, and SUN RGBD show that this approach performs noticeably better than baseline approaches like CLIP2, Uni3D, and UniGS in tasks like open-scene identification, text-3D/image-3D retrieval, and zero-shot categorization.

**Strengths:**

1. The TIGAUSSIAN framework proposed in the article is highly targeted. It tackles the fundamental issues of attribute entanglement encoding and single-view bias in 3D multi-modal alignment tasks by creating multi-modal processing modules like the multi-branch 3DGS Tokenizer. It offers a fresh approach to text-image-3DGS multi-modal alignment, surpassing the drawbacks of current techniques.

2. They rigorously validate the effectiveness of the method, and conducts verification on three major datasets, including Objaverse, by designing four core person scenarios such as zero-shot classification. The baseline approach covers popular 3D representation techniques and uses ablation tests to demonstrate the need for each module design.

3. With a new design, a well-defined experimental procedure, and promises to make the source code publicly available, this approach offers a reproducible reference for future 3D multimodal related work and actively fosters the growth of related disciplines in 3D multimodal research.

**Weaknesses:**

1. The article omits information about the experiment's parameter base and indicators, such as the number of multi-view generations (6) and the 3D-text projection module's parameter sensitivity, which have not been confirmed by tests. Additionally, when it is actually implemented, it does not report the pertinent information (such the inference delay on A100 GPU).

2. It is unclear if baseline models (such UniGS and Duoduo-CLIP) have implemented the same preparation procedures as TIGAUSSIAN (like preprocessing in the downsampling stage) when compared to baseline approaches. The persuasiveness of the outcome is affected by the comparing conditions.

3. Its specific performance in multi-objective, real-world outdoor, and other complicated 3D settings cannot be verified due to the absence of generalization testing in complex scenarios. Additionally, it is not stated how diverse the 3DGS data sources were used in the trials, which makes it impossible to completely illustrate how generalizable the strategy is in arbitrary situations.

**Questions:**

1. How do the issues with the TIGAUSSIAN framework presented in this paper specifically manifest themselves in the generalization testing in complex scenes (like occluded multi-targets and real outdoor scenes) and the rationality verification of important parameters (like the number of Transformer layers in the 3D-text projection module and the number of multi-view generations)? What fixes exist for the issues described above?
2. The experiment's Objaverse, ABO, and SUN RGBD datasets are all openly accessible. Do any self-constructed datasets exist? Will they become open-source if that is the case?

---

> ### Author Response · Authors · 2025-11-27
> **Reply to Reviewer hpbf**
>
> We sincerely appreciate your valuable feedback. All the experiments and discussions below will be included in the final version of the paper, including citations and discussions of relevant baselines.
>
> Concerning weaknesses:
> 1. Considering the **parameter sensitivity**, we've conduct the ablation study on different modules in Sec.4.3, and also perform scaling up experiments in supplementary material Sec.H. As for the **number of multi-view images**, we perform additional ablation study:
>
>     Tab.C. Ablation study on different number of multi-view images.
>     |  Multi-view Number         | Classification Top-1 | Image Retrieval Top-1 |
>     |  ----     | ----       | ----            |
>     | 1 | 35.71  |  40.52  |
>     | 3      | 39.61  |  46.79  |
>     | 6  | **41.76**  |  52.11  |
>     | 12   | 40.52      |  **52.37**  |
>
>     Experimental results have shown that selecting more multi-view images does not necessarily mean better multimodal alignment. We hypothesize that this is due to over-alignment between the image and 3DGS modalities when too many views are used, causing the fused visual features to become overly specific and thus harder to align with the more abstract text representations. In addition, as the number of perspectives increases, the requirement for the authenticity of multiple perspectives will also increase, and the number of model parameters will also multiply. Experimental data shows that selecting 6 multi view images can achieve a balance.
>
> 2. **Details of baselines.** We've described the baseline details in supp. Sec.A.3. Specifically, we use the exactly same data of UniGS to perform fair comparison. As for Duoduo-CLIP, the selected multi-view images were the same as UniGS and TIGaussians. Notice that Duoduo-CLIP does not involve an explicit 3D representation but directly using multi-view representation to store the 3D features.
> 3. **Discussions of limitations and generalizability.** One of the limitations we addressed in the paper is that our method may perform degraded on occluded multi objects or real outdoor scenes. This mainly due to lack of high-quality outdoor GS assets. In addition, feature encoding for large 3DGS scenes is also a challenging task. Moreover, we've stated the 3D-3D feature correlation between heterogenous 3DGS (reconstruced V.S. generated) in supp. Sec.G. The experiment demonstrates that our metod is insensitive to the data source of 3DGS, thus proves the generability of TIGaussians.
>
> Concerning Questions:
> 1. Regarding the **limitations**:
>     1. Occluded Multi-Target Scenarios: In highly occluded multi-object scenes, the current TIGaussian framework may suffer from incomplete Gaussian reconstruction due to limited visible views, leading to fragmented or inaccurate 3D representations. This can degrade cross-modal alignment, especially when text descriptions emphasize occluded parts.
>     2. Real Outdoor Scenes: For real-world outdoor environments with large scale, lighting variations, and dynamic backgrounds, the fixed number of Gaussians and rasterization resolution may limit detail fidelity. The growth and development of related 3D datasets will greatly promote the application of our methods in real-world scenarios.
>
>     The experiments on SUN RGBD dataset can somehow reveal the generalizability of our model, since this is a real-world indoor dataset with some occluded objects. A possible solution for previous issue is training through data augmentation. That said, we acknowledge these limitations and are exploring extensions such as adaptive density control and pose refinement modules to improve robustness in future work.
>     As for some **detailed parameters** of the model design, we conduct experiments on number of selected multi-view images as we presented above. As for the number of Transformer layers in 3D-text projection module, we followed the BLIP-2 configuration.
>
> 2. The experiments described in the paper did not involve any self-constructed or close-source private dataset.

---

### Official Review · Reviewer_BqVr · 2025-11-07

**Soundness:** 3
**Presentation:** 3
**Contribution:** 2
**Rating:** 4
**Confidence:** 3

**Summary:**

This paper proposes TIGaussian, a new text-image-3D contrastive learning paradigm. Based upon the previous work UniGS, it introduces three core modules: (1) decoupled 3DGS tokenizer to avoid entangled 3D representation; (2) mulit-view image fusion to avoid degraded 3D representation; (3) a 3D-text projection module to adapt the 3D latent space for better text-3D alignment. Experiments on Objaverse and ABO show improved performances.

**Strengths:**

1. The paper is well-structured and easy-to-follow. It focuses on improving different designs of current 3DGS-based text-image-3D contrastive method and shows clear improvement.

2. The multi-vew image fusion idea makes intuitive sense to me. Forcing the alignment of a 3D object and a single-view 2D image is clearly suboptimal.

**Weaknesses:**

1. The benchmark comparison is insufficient. Only three baseline methods are listed in Table 1/2. Strong competitors like ULIP [1], OpenShape [2], and MixCon3D[3] should also be included. For example, OpenShape and MixCon3D score a top-1 accuracy of 46.8 and 52.5 on Objaverse-LVIS, which is significantly better than the 41.76 top-1 accuracy of TIGaussian, making one wonder the effectiveness of the proposed approach.

2. Similarly, results on other important benchmarks like ModelNet40 and ScanObjectNN should be reported, as in OpenShape.

3. The image fusion idea is not new. MixCon3D has already shown the effectiveness of aggregating multi-view image features before alignment, and thus make the contribution of this work less novel.


**References**

[1] Xue, Le, et al. "Ulip: Learning a unified representation of language, images, and point clouds for 3d understanding." Proceedings of the IEEE/CVF conference on computer vision and pattern recognition. 2023.

[2] Liu, Minghua, et al. "Openshape: Scaling up 3d shape representation towards open-world understanding." Advances in neural information processing systems 36 (2023): 44860-44879.

[3] Gao, Yipeng, et al. "Sculpting holistic 3d representation in contrastive language-image-3d pre-training." Proceedings of the IEEE/CVF Conference on Computer Vision and Pattern Recognition. 2024.

**Questions:**

1. Why use a pre-trained multi-view diffusion model to generate multi-view images from a single-view perspective? If the data contains multiple text-image-3D triplet of a single object, isn't it more straightforward to fuse the representations of multi-view **real** images? Seems to me the introduction of a diffusion model might also introduce hallucination and might be a potential bottleneck of this method.

2. What is the core difference between 3DGS-based 3D contrastive learning approaches and point-cloud-based or mesh-based approaches? The authors are encouraged to add related discussions and show performance comparisons across different approaches in the paper.

---

> ### Author Response · Authors · 2025-11-27
> **Reply to Reviewer BqVr**
>
> We sincerely appreciate your valuable feedback. All the experiments and discussions below will be included in the final version of the paper, including citations and discussions of relevant baselines.
>
> Concerning weaknesses:
> 1. **More baseline comparison.** We conduct additional baseline comparison on OpenShape, TAMM and MixCon3D and ReCon++. Since we've already tested that ULIP2 performs inferior to our method, we omit the comparison with ULIP. We report the full model performances on Objaverse-LVIS, which is trained on ensembled dataset (without LVIS), with more than 800k objects. We also re-train these models with the same amount of objects (100k) for fair comparison. The results can be found in Tab.A and Tab.B in **General Responses**. As for the best performance reported in MixCon3D with Top-1 accuracy of 52.5, it is trained with the ensembled dataset with Objaverse-LVIS, which means the evaluate dataset is contained in training dataset, which is not suitable for "zero-shot classification". The MixCon3D trained on Ensembled dataset without LVIS reaches 47.5, while our model trained on Objaverse-800k without LVIS reaches 50.1.
> 2. **Concerning about more benchmark comparisons.** We've reported the performance on Objaverse, ABO and SUN RGBD datasets. As for ModelNet and ScanObjectNN dataset, there's currently a lack of high-quality 3DGS reconstruction results and corresponding multi view rendered images.
> 3. **Discussion of MixCon3D.** The key difference between TIGaussian and MixCon3D lies in our proposed 3D encoding scheme for the 3DGS modality. While MixCon3D concatenates multi-view image features with 3D point cloud features to form a holistic object-level 3D representation—followed by projection to obtain a joint embedding—our approach introduces a dedicated and independent 3D encoding process. This allows our model to generate meaningful 3D representations without relying on multi-view images during encoding, effectively decoupling and simplifying the representation learning pipeline. Moreover, we present a 3D-to-3D similarity matrix to simulate the 3D retrieval performance. By TIGaussian, the 3DGS can be directly encoded by our proposed 3DGS tokenizer, without the input of other modals, i.e., text or images.
>
> Concerning questions:
> 1. **Question about multi-view images source.** It's fine to use the pre-rendered images, while we conduct the ablation study in supplementary (see Sec. D.) However, for some dataset without direct multi-view images or when direct inference with single-view image, we can easily leverage the multi-view diffusion prior to generate multi-view image without additional effort. As for the hallucination problems, our experiment in Sec.D. has proved that there's no much difference between generated images and gt images.
> 2. **Discussion of different 3D representations.** Below, we analyze the core differences between 3DGS-based contrastive learning and existing point-cloud/mesh-based approaches from three perspectives: representation expressiveness, encoding efficiency, and cross-modal alignment capability.
>
> + **Comparison with Point-Cloud-Based Methods**:
> Traditional point clouds are typically represented as unordered sets of 3D coordinates, often augmented with color or normal vectors (e.g., 6D [x,c]). While simple and memory-efficient, this representation lacks explicit surface geometry and appearance details. In contrast, 3DGS provides a richer, differentiable representation with per-Gaussian attributes such as position, scale, rotation, opacity, and spherical harmonics (SH) coefficients. This enables more structured and semantically meaningful encoding. To leverage these properties, we design a dedicated 3DGS tokenizer with attribute disentanglement, which explicitly separates geometric and appearance features — an ablation study in Section 4.3 validates its effectiveness.
> + **Comparison with Mesh-Based Methods**:
> Meshs offer dense, topologically connected surfaces and are powerful for high fidelity modeling. However, the storage overhead of mesh is often several times higher than that of sparse point clouds and 3DGS. In addition, the encoding overhead for such complex structures will be higher, especially for high data volume vertices and patches, which makes mesh based alignment more expensive and inference more complex.
> + **Implications for Contrastive Learning**:
> A key challenge in cross-modal contrastive learning is achieving accurate feature alignment across modalities. The explicit and interpretable structure of 3DGS allows us to establish natural correspondences between image regions and 3D primitives (e.g., via visibility or projection), facilitating better cross-modal alignment. On the other hand, the sparse point clouds lack sufficient spatial information, while dense meshes suffer from storage bottlenecks and complex encoding — both hinder effective modality fusion.

---

### Author Response · Authors · 2025-11-27
**General Responses to All Reviewers**

We thank all reviewers for their constructive feedback. Considering the simplicity of the response, we hereby address the questions from reviewr BqVr and yocs regarding more baseline comparisons.

Notice that since most of the baselines leverage an ensembled point cloud dataset (including Objaverse-LVIS evaluation dataset) for training, which has about 876k instances with 10000 points per object, while the results reported in our paper is conducted on only 100k training instances with 1024 points per object. To ensure fair comparison, we reimplement several baselines including Recon, Recon++, TAMM and MixCon3D by training on Objaverse 100k dataset, and also scaling up our TIGaussians to train on Objaverse 800k dataset without Objaverse-LVIS. The comprehensive results demontrate the superior performance of our proposed method, demonstrating the effectiveness of 3DGS representations and the proposed tokenization and alignment strategies.

Tab.A. Comparison of object classification task on Objaverse-LVIS datset.
|           | Base Model | Training Dataset | Sampled Points |   Top-1 | Top-3 | Top-5 |
|  ----     | ----       | ----             |   ----         |   ----  | ----  | ----  |
| ReCon     | PointBERT  |  Objaverse 100k  |   10000        |   25.0  | 45.6  | 50.1  |
| TAMM      | PointBERT  |  Objaverse 100k  |   10000        |   22.7  |  40.1 | 48.5  |
| MixCon3D  | PointBERT  |  Objaverse 100k  |   10000        |   32.3  |  52.5 | 61.5  |
| ReCon++   | ViT-S      |  Objaverse 100k  |   10000        |   26.9  |  47.5 | 53.6  |
| ReCon++   | ViT-B      |  Objaverse 100k  |   10000        |   31.0  |  50.5 | 55.2  |
| Ours      | Uni3D-S    |  Objaverse 100k  |   1024         |   41.8  |  62.7 | 69.1  |
| Ours      | Uni3D-S    |  Objaverse 100k  |   10000        |   **46.7**  |  **68.6** | **74.5**  |
| Ours      | Uni3D-B    |  Objaverse 100k  |   1024         |   *46.6*  |  *67.5* | *75.4*  |


Tab.B. Comparison of object classification task on Objaverse-LVIS datset with larger training dataset.
|           | Base Model | Training Dataset | Sampled Points |   Top-1 | Top-3 | Top-5 |
|  ----     | ----       | ----             |   ----         |   ----  | ----  | ----  |
| OpenShape | PointBERT  |  Ensembled w/o LVIS  |   10000  |   39.1  | 60.8  |  68.9 |
| TAMM      | PointBERT  |  Ensembled w/o LVIS  |   10000   |  42.0  |  63.6 | 71.7  |
| MixCon3D  | PointBERT  |  Ensembled w/o LVIS  |   10000   |   47.5  |  69.6 | 76.2  |
| ReCon++   | ViT-B      |  Ensembled w/o LVIS  |   10000   |   *49.6*  |  *70.2* | *78.4*  |
| UniGS     | Uni3D-S    |  Objaverse 800k w/o LVIS |   1024  |   48.8  |  69.1 | 76.9  |
| Ours      | Uni3D-S    |  Objaverse 800k w/o LVIS |   1024  |   **50.1**  |  **73.6** | **79.6**  |

All rebuttal discussions and supplementary experiments will be added to the revised version of the paper, and more citations will be added. We would like to express our gratitude once again to all the reviewers for their efforts.

---

### Author Response · Authors · 2025-11-29
**Summary Reply to AC and all Reviewers**

We've been noticed by the ICLR Program Chairs of the the recent security incident involving reviewer anonymity on the OpenReview platform. We fully respect the integrity of the peer-review process and appreciate the efforts made by the organizers to maintain fairness under such circumstances. Throughout the process, we have strictly adhered to all guidelines and refrained from accessing any non-public information. Despite the closure of the reviewer response & discussion phase, we would like to formally summarize the key points addressed in our original responses to ensure that the technical merits of our work remain clearly communicated.

# Summary of Key Responses to Reviewer Concerns
1. **Comprehensive Baseline Comparisons & Fair Evaluation**. (Reviwer BqVr & yocs)
* We re-implemented several baselines using a fair setting: trained on the same Objaverse-100k dataset, matching our data scale.
* We also scaled up our model to train on Objaverse-800k (w/o LVIS) for direct comparison with ensemble-trained models.
* As shown in Tab. A & B, our best model achieves Top-1 accuracy of 50.1, outperforming MixCon3D (47.5) and ReCon++ (49.6), both under zero-shot evaluation conditions.

    This demonstrates that our performance gain is not due to data contamination or unfair training advantages.

2. **Clarification of Methodological Advantages over Prior Work**. (Reviewer BqVr & Xq4U)

* vs. MixCon3D/JM3D:
Unlike prior works that concatenate image and point cloud features early (MixCon3D) or rely on indirect pre-alignment via text-image priors (JM3D), we propose an independent 3DGS encoder with a dedicated 3DGS tokenizer that enables direct, fine-grained 3D–text alignment. This decouples 3D representation learning from visual priors, improving generalization.

* vs. Point Cloud / Mesh Representations:
We clarified that 3DGS offers richer geometric and appearance cues (position, scale, rotation, SH coefficients) compared to raw point clouds, while being more memory-efficient than meshes. Our ablation (Sec.4.3) validates the effectiveness of attribute disentanglement in the proposed tokenizer.

    These discussions were supported by new analysis in Section 4.3 and supplementary materials.

3. **More Ablation Studies**. (Reviwer hpbf & yocs)
* Number of Multi-View Images:
We conducted ablation studies (Tab. C) showing that that 6 views achieve optimal balance between coverage and modality coherence.

* Scalability:
Experiments on Objaverse-800k show consistent gains (Top-1 from 46.6 → 50.1), proving strong scalability without relying on external ensembled datasets.
* Impact of Pre-trained Point Cloud Encoders:
The position and color attribute of 3DGS is similar to point cloud, thus the spatial tokenizer could directly benefit from the pre-trained point cloud encoder (e.g., Uni3D with ViT). Besides, the generalizability of pre-trained point cloud encoder could be inherited and make the training process easier. Our architecture specifically exploits the full richness of 3DGS attributes (beyond position/color), validated through ablation and extension experiments (e.g., unfreezing backbone).

4. **Computational Efficiency and Practical Feasibility**. (Reviwer BqVr & Xq4U & hpbf & yocs)
* The total GFLOPs of our model is manageable (~113M), with the 3DGS Tokenizer accounting for ~88.2M.
* Details and comparisons of inference time cost and the source of multi-view images are discussed in the *supplementary materials* (Sec.D&E).
* All experiments use publicly available datasets; no private or closed-source data was involved.

# Summary of Discussions
Our method, TIGaussians, introduces a novel 3D-aware representation learning framework that leverages 3DGS as a rich 3D primitive for cross-modal alignment. We propose a dedicated 3DGS tokenizer with attribute disentanglement to fully exploit the geometric and appearance cues in 3DGS. Furthermore, the 3D-aware image feature fusion and 3D-text projection module also contribute to the effective modal alignment. Extensive experiments show that our approach achieves state-of-the-art performance on zero-shot 3D classification and retrieval tasks, outperforming strong baselines such as UniGS and ReCon++ by clear margins. Ablation studies validate the effectiveness of key components, demonstrating the performance gains in 3D representation learning.

To conclude, we have dedicated to addressed all major concerns raised by the reviewers—including baseline reproducibility, architectural justification, parameter sensitivity, and generalization limitations—through extensive experiments and clarifications. All rebuttal content, including additional tables, citations, and expanded discussion, will be integrated into the final revision.

We sincerely thank all reviewers and the ACs for their time, insightful feedback, and dedication to scientific rigor. Despite the unfortunate platform issues, we remain committed to transparent and constructive scholarly communication.

---

### Meta-Review · Area_Chair_RC74 · 2026-01-05

**Summary:**

The reviewers had mostly positive scores before the rebuttal. The main points raised by reviewers were the following. (a) Two reviewers were particularly concerned about baselines and fairness, noting the absence of recent datasets, as well as concerns with how some of the baselines were conducted. (b) There was a question about scalability and computational costs, noting that the method didn't train on the full dataset, and two reviewers asked for more complex datasets. (c) There was some questions about the method and justification of design choices, particularly why the method should use diffusion if a 3D model is available.

**Reviewer Concerns:**

The authors rebuttal strongly addressed the concerns about baselines, and show results with the same dataset for fairness. They also scaled up the method to larger datasets, showing performance gains. While they couldn't generalize to outdoor scenes quite yet, this is not a major limitation. The authors provided information about the computational cost, as well as answered methodological questions particularly about why diffusion should be used when a 3D model is available. They also answered the concern about single image versus multiple image.

**Reviewer Scores:**

The reviewers were initially positive, and I expect the new experimental results would have further convinced the reviewers. While there are some limitations in the experiments (for example the lack of outdoors), the new results make a strong case. As the AC, I also examined the paper closely and found the problem setup interesting.

---

### Decision · Program_Chairs · 2026-01-26

Accept (Poster)